# The Effect of Rare Earth Metals Alloying on the Internal Quality of Industrially Produced Heavy Steel Forgings

**DOI:** 10.3390/ma14185160

**Published:** 2021-09-08

**Authors:** Petr Jonšta, Zdeněk Jonšta, Silvie Brožová, Manuela Ingaldi, Jacek Pietraszek, Dorota Klimecka-Tatar

**Affiliations:** 1Faculty of Materials Science and Technology, VŠB—Technical University of Ostrava, 708 00 Ostrava, Czech Republic; petr.jonsta@vsb.cz (P.J.); zdenek.jonsta@vsb.cz (Z.J.); silvie.brozova@vsb.cz (S.B.); 2Faculty of Management, Czestochowa University of Technology, 42-201 Czestochowa, Poland; d.klimecka-tatar@pcz.pl; 3Faculty of Mechanical Engineering, Cracow University of Technology, 31-155 Cracow, Poland; jacek.pietraszek@mech.pk.edu.pl

**Keywords:** heavy forgings, low-alloyed structural steels, rare earth metals (REMs), nonmetallic inclusions, structure, mechanical properties, industrial quality of material, special processes

## Abstract

The paper presented the findings obtained by industrial research and experimental development on the use of rare earth metals (REMs) in the production of heavy steel ingots and their impact on the internal quality of the 42CrMo4 grade steel forging. REMs alloying was carried out after vacuuming the steel. A relatively large melting loss of cerium (about 50%) and its further decrease in casting due to reoxidation were observed. Refinement of structure and better mechanical properties of forged bar containing about 0.02 wt.% of Ce compared to that of the standard production were not achieved. The wind power shaft with content of about 0.06 wt.% of Ce showed high amount of REM inclusions, which were locally chained, and in some cases, initiated cracks. Four stoichiometrically different types of REM inclusions were detected in forgings, namely (La-Ce)_2_O_2_S + (La-Ce)O_2_ + SiO_2_ (minority); oxygen, phosphorus, arsenic, and antimony bound to lanthanum and cerium probably bonded with iron oxides La + Ce, MgO, Al_2_O_3_ a SiO_2_; (La-Ce)_2_O_2_S, FeO, SiO_2_, and CaO or CaS.

## 1. Introduction

Supervision of the quality of products in metallurgical production imposes on the company the obligation to comply with numerous standards and procedures—which results from the specificity of metallurgical processes determined by thermal processes and chemical dependencies of components. Metallurgical processes such as casting, forging, and rolling, etc., are classified as special processes. Therefore, management of the products quality requires a very thorough insight into the reactions and relationships between the treatment processes and the chemical composition of materials. Only compliance with all system principles in the process will allow to obtain products with appropriate functional properties. Qualitative supervision is particularly difficult for such processes, as it is based not on 100% control, but on complex analyses of statistical production control, as in other processes in the field of materials science [1,2,3,4,5,6,7,8,9].

Large steel products produced by complex open-die forging processes are widely used in industries with high quality demands on structural and mechanical properties. One of the key fields is undoubtedly power engineering, especially renewable sources, where heavy and long forgings are used, for example, for rotor shafts of wind turbines [10,11,12,13,14]. The input semifinished product for the production of these shafts is heavy steel ingots, in which there may be problems with achieving the required internal quality due to the occurrence of a very inhomogeneous and coarse-grained structure [15,16,17,18].

For the given products (forgings), steels alloyed with chromium, nickel, molybdenum, or steels microalloyed with vanadium are used. These are mostly 42CrMo4, 34CrNiMo6, 10CrMo9-10, 55NiCrMoV7, and 27NiCrMoV15-6 steel grades. These steel grades require very low sulfur contents (up to 0.005 wt%, or 0.002 wt.% for the 27NiCrMoV15-6 grade) and require high cleanliness [19,20,21,22]. With the growing demands of customers on the mechanical properties of forgings, it is also necessary to develop new types of steel, also using nontraditional elements that were not previously used in common metallurgical practice. These elements include rare earth metals (REMs), most commonly, cerium and lanthanum, which are used to modify nonmetallic inclusions (shape control), deep purifying (deoxidation, desulphurisation, removal or reduction of elements such as P, S, As, Sb, Pb), and alloying [23,24,25,26,27,28].

The deoxidizing properties of cerium and lanthanum are significantly higher than that of aluminum, which is used as the strongest deoxidizing agent in conventional (structural) steels. Cerium and lanthanum have a significantly higher affinity for oxygen than other common deoxidizing elements, such as manganese and silicon. These two most frequently used REM elements generally form very stable oxides, Ce_2_O_3_ and La_2_O_3_, respectively. REM are also characterized by a high affinity for sulfur and form sulfides CeS, Ce_2_S_3_, and La_2_S_3_, or oxysulfides (RE)_2_O_2_S, which is reflected in their use by a reduction in the proportion of MnS and a better degree of desulfurization of steel. The advantage of using rare earth metals is the even distribution of sulfur in the steel even during slow cooling, which is due to the high melting point of most REM oxides and sulfides. The melting point of these inclusions is higher than the solidification temperature of the steel, which leads to the inclusions being precipitated earlier and being located inside the steel crystals [29,30,31]. The average length of sulfide inclusions decreases sharply with increasing cerium content in steel, up to a content of at least about 0.06 wt.% Ce. Then, it remains almost constant. Cerium also forms other stable compounds in steel, namely carbides and nitrides. Knowledge of the thermodynamic properties of liquid and solid phases at high pressures and high temperatures is necessary for understanding melting phase relations in complex geological systems. A large amount of literature on this aspect is available; recently, the thermochemical evolution and thermodynamic behavior of oxide inclusions in specific steel compositions were presented by Belmonte et al. [32,33]. The elements of the REMs have a very high ability to deoxidize and desulfurize steel (their effect is definitely stronger than the addition of magnesium), which results in minimizing the amount of oxygen and sulfur. High reactivity of REMs causes the formation of oxides and sulfides, the presence of which may additionally modify the composition and morphology of magnesium oxides and sulfides [23,34].

REMs refine the casting structure of the steel, reduce the zone of columnar dendrites, and improve the quality of the macrostructure of cast ingots, which also has a positive effect on the mechanical properties of steel products [35,36].

When using REMs, microsegregations of alloying elements, especially silicon, phosphorus, and sulfur, are limited. Furthermore, the presence of cerium in steel has a favorable effect on its recrystallisation. REM alloying also increases fatigue strength. Fatigue cracks are reduced due to the formation of sulfide inclusions of a favorable nature [37].

The amount of REMs required to achieve the optimum content in the steel, especially in terms of mechanical properties, depends on the oxygen and sulfur contents in the steel before their alloying [38].

To ensure high utilization of cerium when alloying into liquid steel, it is recommended to inject the filled profile with mischmetal (typically approx. 50% cerium, 25% lanthanum with smaller amounts of neodymium and praseodymium) after vacuum degassing, i.e., after reducing the content of oxygen, sulfur, and inclusions in the steel. However, sufficient time must be provided for the formation of nonmetallic inclusions (REM oxides and sulfides, which are formed due to the high affinity of rare earth metal elements for oxygen and sulfur) after alloying with mischmetal into liquid steel [39].

Appelberg et al. [40] discussed the effect of mischmetal on the behavior of inclusions in molten stainless steel. During the use of the elements Ce, La, Al, O, clogging of the immersion nozzles occurs during steel casting with oxides of these elements. There was an increased tendency for large inclusions to form, resulting in clusters when aluminum was used, as previously reported by Yin et al. [41,42]. Nakajima a Mizoguchi [43] presented theoretical analyses and confirmed that the capillary interaction is strongly influenced by particle shape, size, and surface tension of inclusions in the melt. Kojola et al. [44] studied nozzle clogging mechanisms during casting of REM treated stainless steels weighing of 350 kg. Experimental results showed differences between fast and slow clogging rates. Steel containing mainly small single inclusions clogged faster than steel containing mainly large inclusion clusters. The reason was believed to be that the small inclusions could stick to the nozzle wall at narrow passages where the steel flow velocity was high, while the larger ones could not. The source of the small inclusions was believed to be reoxidation.

The effects of the interaction between coating layer and the submerged entry nozzle refractory materials on clogging of REM alloyed stainless steels were studied by Mamarpour et al. [45]. The study showed the vital effect of glass/silicon powder coating on the clogging mechanisms. It is suggested to use a low alkali free glaze such as borosilicate and E glasses to prevent its harmful effects on clogging phenomenon.

The addition of rare earth (RE) to steel also affects the corrosion resistance of steel [46,47,48]. The REM addition reduced the surface electrochemical activity and the adsorption tendency of Cl^−^ on the metal surface [47]. The addition of rare earth metals has a significant effect on both metastable and stable pitting processes [46]—the dissolution of inclusions induced the electrochemical activities of pitting corrosion and further inhibited the propagation of local corrosion [48]. Yang et al. [49] studied the effect of cerium on the corrosion resistance and mechanical properties of A36 steel sheet in a seawater environment (3.5% NaCl solution). Cerium was added to the steel during production. The experimental results showed that the corrosion resistance of A36 steel containing cerium was higher compared to that of steel A36, which did not contain cerium. Mechanical properties were also enhanced; tensile strength increased by 6%, and yield strength by 8%. At a cerium content of 0.009 wt.% and at a test temperature of 0 °C, the notch strength of A36 steel increased by 9%.

Adabavazeh et al. [50] studied the effect of different cerium contents on the microstructure and morphology of inclusions in SS400 steel (A36). A sample with an optimal amount of 0.0235 wt.% cerium and a S/O ratio of about 7 contained predominantly cerium oxides, oxysulfides, and only a small amount of sulfides. By increasing the amount of cerium and decreasing the S/O ratio, an increased amount of oxysulfide inclusions was revealed. Inclusions of 4–7 µm in size served as heterogeneous nucleation sites for intra-granular acicular ferrite formation.

The influence of Ce/La on the microstructure of cold work tool steel is presented in [51]. About 0.03 wt.% mischmetal was added. The samples were heat-treated at 1130 °C/3 h/slow cooling in the furnace to 700 °C/2 h/slow cooling in the furnace to room temperature. The results showed that after modification with Ce/La, the morphology, size, and distribution of Cr_7_C_3_ carbides change greatly. The carbides network tends to break, and all carbides are refined and distributed homogenously in the matrix; they also reduce the size of chromium carbides and increase the dissolution of carbides during heat treatment. The results of mechanical tests showed that the ductility of the steel increased by 75% without decreasing the hardness.

Huang et al. [52] studied the microstructure and mechanical properties of HSLA-D6AC steel with Ce content. The results of the experiment showed that the addition of a certain amount of Ce could refine grains and martensitic laths, as well as refine VC precipitates, which increases not only the strength of the steel but also its toughness and plasticity. The morphology of martensite also changed from twin to dislocation martensite. After the addition of Ce to the D6AC steel, the impact strength increased by about 50 J, and the originally quasifission areas on the fracture surfaces were transformed into ductile areas. The authors state that excellent complex mechanical properties were achieved.

The influence of REM on the structural properties and impact strength of H13 ingot steel was discussed by Gao et al. [53]. The best results were obtained with a REM content of 0.015 wt.% when 90% of inclusions smaller than 2 µm were detected in the steel matrix.

The study of the effect of 0.068 wt.% cerium on plastic properties of SA-508 steel for heat treatment is given in [54]. Microstructural analysis showed that cerium segregation grain boundary and a finer-grained structure contribute to increasing the plastic properties of this steel at high temperatures (above 700 °C).

A large amount of knowledge concerning the influence of REMs on a wide range of properties, especially of high alloy steels or low carbon steels [55,56,57,58], is presented in the technical literature, but comprehensive research in the field of structural steels is still needed. Moreover, these are mostly applications performed in laboratories or pilot plants, the conditions of which differ more or less from the real conditions at heavy metallurgical plants.

The aim of the paper is industrial research and experimental development of the use of REMs in the production of heavy steel ingots. The effect of REMs alloying, specifically cerium on the microcleanness and refinement of microstructure and mechanical properties of forgings made of 42CrMo4 steel, was studied.

The solution to the problem presented in this paper is not yet openly published by any other authors. REMs alloying of heavy ingots, made of structural steel, tens of tons in weight, is not commonly available in technical literature. Therefore, the results obtained are novel and can be useful for other researchers.

## 2. Materials and Methods

Operational experiments were designed on the basis of available literature and the results of our previous experimental activities [59,60,61,62,63]. Experiments were realized in two stages, when two trial heats (marked A, B) of low-alloy structural Cr-Mo steels, which were alloyed by REMs in the form of Mischmetal with a composition of 59.3 wt.% of Ce, 36.5 wt.% of La, 0.2 wt.% of Fe, and 0.4 wt.% of Mg, were gradually produced in the electrical steel plant. The final cerium content in the steels was targeted at 0.05–0.07 wt.%. Verification of the influence of REMs on the properties of cast steels was performed on both the ingot (heat A) and forgings (heat A, B).

### 2.1. Research Designed for the Low-Alloyed Structural Cr-Mo Steel, Heat A

Two ingots, each weighing about 23 t, were cast from heat A and control cooled. One of the ingots was flame cut longitudinally in the axis, and then the heat-treated layer was milled to a depth of about 40 mm to perform chemical analyses to determine the distribution of cerium along the cross-section of the ingot and verify its effect on segregation of selected elements, especially carbon and sulfur. Chemical analyses were performed on an ¼ ingot (35 analyses each, of which 5 on the head, 25 on the body and five on the bottom) using mobile spectrometer RTG NITON XL2 (ThermoFisher SCIENTIFIC, Carlsbad, CA, USA) directly on the ingot, and also on samples taken from ingot by drilling (chips, core drilled samples), which were analyzed using stationary RTG X-lab devices (SPECTRO Analytical Instruments, GmbH, Kleve, Germany) and GDS 850 (LECO Instrumente, GmbH, Mönchengladbach, Germany).

A bar with a diameter of 540 mm was forged from the second ingot of heat A. The forging reduction was 5. After forging, the bar was cooled in air and subsequently quality heat-treated by regime 840 °C/14 h/water + 620 °C/16 h/air. After descaling the bar, ultrasonic testing according to EN 10 228-3 [64] was performed with a DIO 1000PA (STARMANS electronics, s.r.o., Prague, Czech Republic). To verify the chemical composition, microcleanliness, microstructure, and mechanical properties of the forging, a test segment with a thickness of approximately 60 mm was taken from the front of the bar (from the head of the original ingot). Samples for individual analyses were always prepared from the subsurface area of the segment (forging), ¼ diameter and center. The control chemical analysis of forging samples was verified by optical emission spectrometry on a stationary SPECTROMAX analyzer (SPECTRO Analytical Instruments, GmbH, Kleve, Germany), in the case of oxygen, nitrogen, carbon, and sulfur determination by elemental analysis on a stationary combustion element analyzer Eltra ONH 2000, or Eltra CS2000 (ELTRA, GmbH, Haan, Germany). The phosphorus content was determined by GDOES “Bulk” analysis using a GDA750A spectrometer (SPECTRUMA Analytic, GmbH, Hof, Germany).

The content of nonmetallic inclusions was determined according to ASTM E45, method A, at 100× magnification. The size of the original austenitic grain was determined after heat treatment according to ASTM E 112, using the comparative method, at 200× magnification.

Structural analyses were performed after etching samples in a solution of 4% HNO_3_ in ethyl alcohol. IX 70 and GX 51 light microscopes (OLYMPUS Co., Tokyo, Japan) were used for microscopic observations.

Electron microscopy analyses (EDAX, EDS) were performed using a JSM-6490LV scanning electron microscope (JEOL Co., Tokyo, Japan) equipped with an x-act INCA energy dispersion spectrum analyzer (Oxford Instruments, Inc., Abingdon, UK).

The mechanical properties of the forging were verified by a tensile test using a 100 kN universal testing machine (walter + bai ag, Löhningen, Switzerland) and by Charpy V impact test on a Charpy PH300 (walter + bai ag, Löhningen, Switzerland).

Test samples for mechanical tests were taken in the transverse direction from the forging segment, always from the area below the surface, ¼ diameter and center. From each area, 3 test specimens for the tensile test and 3 test specimens for the Charpy impact test were prepared in each case. The test specimens for the tensile test had a diameter of 10 mm and an initial gauge length of 50 mm. Standard test specimens with a V-notch with a depth of 2 mm were used for the Charpy impact test. The test was always performed at ambient temperature.

### 2.2. Research Designed for the Low Alloyed Structural Cr-Mo Steel, Heat B

One ingot weighing about 50 t was cast from heat B. It was control cooled, and subsequently, the wind power shaft was forged from it. After forging, the shaft was cooled in air and quality heat-treated by the regime of 840 °C/15 h/water + 610 °C/19 h/air. After descaling the forging, ultrasonic testing was performed according to EN 10 228-3 with a DIO 1000PA defectoscope (STARMANS electronics, s.r.o., Prague, Czech Republic). Unacceptable indications were detected on the forging; they were subjected to chemical and structural analyses to clarify the cause of their formation. Similar testing methods were used on the same test devices as in the case of forging made from heat A.

## 3. Results

### 3.1. Results of the Effect of REMs on the Formed Inclusions, Microstructural and Mechanical Properties of Low-Alloyed Structural Cr-Mo Steel Made from Heat A

The production of steel (heat A) was conducted by oxidative melting technology using the technological residue in EAF (Electric Arc Furnace) and subsequently processed on secondary metallurgy equipment LF + VD (Ladle Furnace and Vacuum Degassing). After the evacuation, steel samples, slags were taken, the hydrogen content was measured by HYDRIS^®^ [35], and Ca-Si (0.2 kg/t) was added in the form of a profile. The melt temperature of the steel was 1596 °C; the hydrogen content was 0.000094 wt.%). The chemical composition of the steel after vacuum degassing (before the addition of REMs) and after the addition of REMs is given in Table 1. The analysis of slag after vacuum degassing is given in Table 2. After sampling for chemical analyses, the ladle was driven back into the VD, and after creating a vacuum, REMs in the form of mischmetal were added into the steel in the amount of 70 kg, which corresponded to the assumed final cerium content in the steel of 0.05–0.07 wt.%. The course of alloying with mischmetal pieces was calm; there was no visible reaction of the mischmetal with steel, which was also proved by a camera placed on the lid of the vacuum degassing chamber. Alloying with mischmetal lasted about 5 minutes, and after its completion, the steel was briefly bubbled with argon, taken out of the vacuum degassing chamber, and the steel temperature, oxygen activity, and hydrogen content were measured. CELOX^®^ [36] and HYDRIS^®^ were used to measure the oxygen, respectively hydrogen activity. Subsequently, steel and slag samples were taken.

The temperature drop after alloying with mischmetal was 23 °C, i.e., about 2.3 °C/min. In the case of oxygen activity, its value decreased from the original 0.000385 wt.% to 0.000120 wt.%, i.e., a decrease by 0.000265 wt.%. The melting loss of cerium in the melt was 56.3%, which is also confirmed by the analysis of cerium from the laboratory (0.035 wt.%). Interestingly, there was a decrease in cerium content in steel during casting, with a total decrease of 0.009 wt.%, which corresponds to a melting loss of about 26% compared to that of the melting analysis, suggesting additional reoxidation of cerium during casting with atmospheric oxygen. The cerium melting losses ranged from 50–56% during alloying and VD, and it is also necessary to consider the decrease in the cerium content during casting. If it is required to alloy other similar melts with mischmetal targeting the cerium content in the melt of 0.05–0.07 wt.%, it will be necessary to increase the amount of mischmetal used, which will include higher losses due to melting loss and reoxidation. There was no significant decrease in the contents of Mn and Si; the content of Al decreased only by 0.004 wt.%. Thus, Ce had a favorable effect on the melting losses of these elements.

Slag sampling was performed in three stages of steel production. First, at the beginning of steel ladle processing (LF), then after vacuum degassing, and finally, after the addition of cerium in the form of mischmetal. No significant changes in the composition of slag (CaO, SiO_2_, Al_2_O_3_, MgO) occurred, only the higher content of FeO after alloying with mischmetal was analyzed. This is a fundamental difference compared to that of the experiment in [31], where differences in SiO_2_ and MgO contents were noted, which were explained by the use of a filled profile with cerium containing 33% Si, the age of the ladle, and thus, its wear (MgO).

The cerium content of the slag was also analyzed. Prior to mischmetal alloying, the Ce content in the slag was 0.10 wt.%, after alloying 1.0 wt.%. It is not clear why this happened, and it is surprising; additional analysis would be very useful. According to the calculation, 80 kg of cerium was added to the melt (70 kg was a fact). Therefore, the cerium melting loss was 13.8 wt.%, and cerium utilization would be 86.1%. If this calculation were correct, the cerium content in the steel should be 0.078 wt.%, which, due to the melting analysis of 0.035 wt.%, was not proved. This fact, therefore, indicates a higher melting loss of cerium, namely 50–56%, as mentioned above.

From the experimental heat A, two pieces of forging ingots were cast, each weighing about 23 t. A teeming ladle with a larger diameter of ladle nozzle (90 mm) than in that of [31] was chosen for casting to avoid clogging [15,16,17]. To prevent reoxidation of the steel by atmospheric oxygen, the casting stream was protected with argon. Despite this type of protection, however, partial oxidation of cerium occurred. The casting process was standard, with no signs of problems. The ingot heads were treated with exothermic and insulating powder. The quality of the ingot heads was very good, without significant shrinkage. The ingot surfaces were free of obvious defects.

The performed analyses of the cerium content in the cross-section of the ingot from heat A (Figure 1) show its uneven distribution (Figure 2). The highest cerium content was found in the axial part of the head and in the subhead region of the ingot and was around 0.042 wt.%. Towards the bottom of the ingot, its content then gradually decreased. At the ingot wall, the cerium content was around 0.032 wt.%, and in the bottom area, it ranged from 0.027–0.031 wt.%

The carbon concentration was highest in the subhead area of the ingot in the axial sections. The cross-sectional distribution was more or less uniform, which was similar in the case of sulfur. The hydrogen content was 0.00010 wt.%, 0.000143 wt.%, 0.000127 wt.%, and 0.000132 wt.%.

On the forged bar from the second ingot of heat A, point indications with a size of max 3 mm, clusters of indications and elongated indications with a length of max 2 mm were detected by ultrasonic test.

This was followed by a control chemical analysis, the results of which are shown in Table 3. The values given were always determined by the average of three measurements. The measured contents of the key elements for a given steel grade corresponded to the melting chemical composition; see Table 1. The determined cerium content ranged from 0.023–0.028 wt.%, and thus, a further decrease was recorded compared to that of the melting analysis given in Table 1.

The microcleanliness of all studied samples was very good, with occasional inclusions of type B and D inclusions. In rare cases, type D inclusions with a diameter of about 20 µm were found. Examples of inclusions found are shown in Figure 3a–c; EDS microanalysis of inclusions and area chemical analyses of the sample are in Table 4.

The detected inclusions in the steel matrix were statistically analyzed. Based on the achieved results, four types of inclusions were found, see Table 5. The first group (No. 1) consists of oxide-sulfide inclusions bound to lanthanum and cerium with stoichiometric composition (La-Ce)_2_O_2_S + (La-Ce)O_2_ + SiO_2_(minority).

The second group of inclusions (No. 2) is stoichiometrically complex. Oxygen, phosphorus, arsenic, and antimony bound to lanthanum and cerium were detected. There is probably a bond with iron (Fe content was about 20%).

The third group of inclusions (No. 3) are oxides La + Ce, MgO, Al_2_O_3_ a SiO_2_. The presence of sulfur or phosphorus was not detected.

The fourth group of inclusions (No. 4) is similar to the first group. Stoichiometrically, it corresponds to the complex of phases formed by the compounds (La-Ce)_2_O_2_S, FeO, SiO_2_ a CaO or CaS. Interestingly, neither chromium nor manganese was identified at any stage.

The microstructure was evaluated on transverse cuts of samples, and in all monitored areas (subsurface, ¼ of diameter, the center of forging), it was very similar; uneven, formed by blocks of bainite and ferrite grains and perlite nodules (see Figure 4). The size of the original austenitic grain was the same in the whole cross-section of the forging, G = 9.

The results of tensile tests (average of three tests) and Charpy impact test are given in Table 6, which shows relatively high values of the tensile strength for a given quality and type of the forging while maintaining excellent plastic properties. As expected, all stress characteristics decreased from the surface towards the center of the forging. The achieved values of impact energy seem to be relatively low, even though the microcleanliness of the steel was at an excellent level, as stated above.

### 3.2. Results of the Effect of REMs on the Formed of Inclusions and Microstructural Properties of Low-Alloyed Structural Cr-Mo Steel Made from Heat B

Based on the findings presented in chapter 3.1, experimental heat B was produced. The course of the production at the steel plant was similar to that of heat A. The main difference was the increase of alloying with mischmetal to 100 kg, which was to ensure the final cerium content in the steel in the range of 0.05–0.07 wt.%, as well as the use of higher quality casting ceramics and certain modifications concerning the Ar protection of the casting stream of the steel to minimize the secondary reoxidation of the steel by atmospheric oxygen. The chemical composition of heat B before and after the addition of mischmetal to VD is shown in Table 7. Vacuum degassing of the steel lasted 33 min, bubbling the melt by argon 22 min. The reaction of REMs with Al_2_O_3_ formed clusters during casting [14,15], which resulted in a narrowing of the casting stream. The optimum casting speed was achieved only from the body/head interface of the ingot.

The wind power shaft was forged from a cast ingot weighing approximately 50 t. Unfortunately, during the ultrasonic inspection of the shaft, unacceptable indications were detected; the area of their occurrence is marked in red in Figure 5. This was a frequent occurrence of point indications up to the equivalent diameter of approximately 5 mm, and three elongated unsatisfactory indications. To determine the cause of these unacceptable indications, test segments A, B (see Figure 6) were taken from the “worst” areas of the shaft, from which samples 34/A, 34/B were prepared for chemical and structural analyses.

The control chemical analysis confirmed that the content of all elements mentioned above is in accordance with the melting chemical composition (Table 7) except for the carbon, which showed a slightly higher content, see Table 8.

Based on the visual inspection of the products carried out in accordance with the guidelines of the quality management principles [65], the crack about of 4.5 mm in length was observed in the metal matrix of sample 34/A (Figure 7). In the vicinity and in the filling of considerably coarse and open crack, mostly row-oriented, nonmetallic inclusions were observed, or clusters of nonmetallic inclusions with several contrasts (Figure 8). Point nonmetallic inclusions were present in the metal matrix, except for the crack (Figure 9).

X-ray microanalysis showed that the chemical composition of various nonmetallic inclusions corresponds to complex oxides, or lanthanum and cerium oxysulphides (Figure 10).

For sample 34/B, the observed crack was significantly smaller and finer (Figure 11a) than for that of sample 34/A (Figure 7) and was about 1.7 mm in length. Nonmetallic inclusions in sample 34/B were observed both in its filling (Figure 11b) and in the matrix. Locally, these inclusions interconnected, and small chains were formed (Figure 11c). The frequency of inclusions was also high in sample 34/B.

The identification of the chemical composition of both bright white and slightly grey inclusions proved that these are complex oxides based on lanthanum and cerium with variable amounts of calcium and silicon, or lanthanum and cerium oxysulphides, see Figure 12.

The microstructure of both studied samples 34/A, 34/B was identical, quenched and tempered with the occurrence of segregation bands of darker contrast, guided in the direction of material formation. Perlite, locally small islands of ferrite, were observed in smaller amounts in the metal matrix. A higher frequency of precipitate was found in the segregation. No decarburization was observed around the cracks. Clear photo documentation is shown in Figure 13.

## 4. Conclusions

The paper presented the findings obtained by industrial research and experimental development of the use of REMs in the production of heavy steel ingots and their impact on the internal quality of the 42CrMo4 grade steel forgings.

The most important aspects can be summarized as follows:To ensure high utilization of cerium in the industrial conditions of the electrical steel plant, it is optimal to alloy with mischmetal after vacuuming the steel. However, sufficient time must be provided for the formed nonmetallic REM inclusions to float out after alloying with mischmetal.When alloying with mischmetal in a vacuum degassing chamber, it is necessary to consider a relatively large melting loss of cerium (about 50%). A further decrease should also be considered in casting due to reoxidation.Compared to that of the standard industrial production of the same Cr-Mo steel grade, it is necessary to slightly extend the vacuum time of the steel and, after alloying with mischmetal, to homogenize the steel melt again with a stream of argon of moderate-intensity.Alloying with mischmetal in piece form (Ce and La alloy) does not affect significant changes in the composition of slag, which is a fundamental difference compared to the use of mischmetal in the form of a filled profile with a content of about 30% Si.Refinement of structure and better mechanical properties of forged bar containing about 0.02 wt.% of Ce compared to that of the standard production were not achieved.The wind power shaft with content of about 0.06 wt.% of Ce showed high amount of REM inclusions, locally were chained, and in some cases, initiated cracks.Four types of inclusions in forgings were found:
○Oxide-sulfide inclusions bound to lanthanum and cerium with stoichiometric composition (La-Ce)_2_O_2_S + (La-Ce)O_2_ + SiO_2_(minority).○Stoichiometrically complex; oxygen, phosphorus, arsenic, and antimony bound to lanthanum and cerium were detected. There is probably a bond with iron (Fe content was about 20%).○Oxides La + Ce, MgO, Al_2_O_3_ a SiO_2_. The presence of sulfur or phosphorus was not detected.○Stoichiometrically, it corresponds to the complex of phases formed by the compounds (La-Ce)_2_O_2_S, FeO, SiO_2_ a CaO or CaS. Interestingly, neither chromium nor manganese was identified at any stage.


The achieved results can certainly contribute to the further development of modern structural steels, which are used for components working in demanding conditions and are subject to high demands in terms of structure and mechanical properties.

## Figures and Tables

**Figure 1 materials-14-05160-f001:**
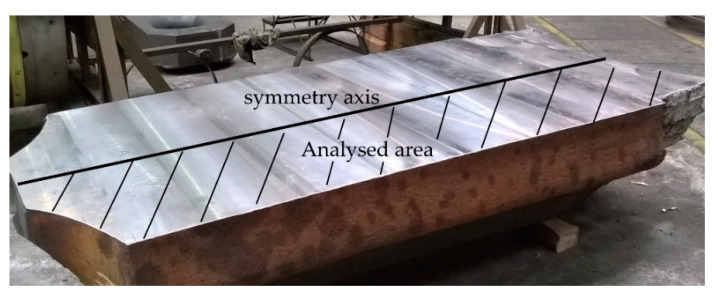
Half of experimental ingot ready to perform chemical analysis (heat A).

**Figure 2 materials-14-05160-f002:**
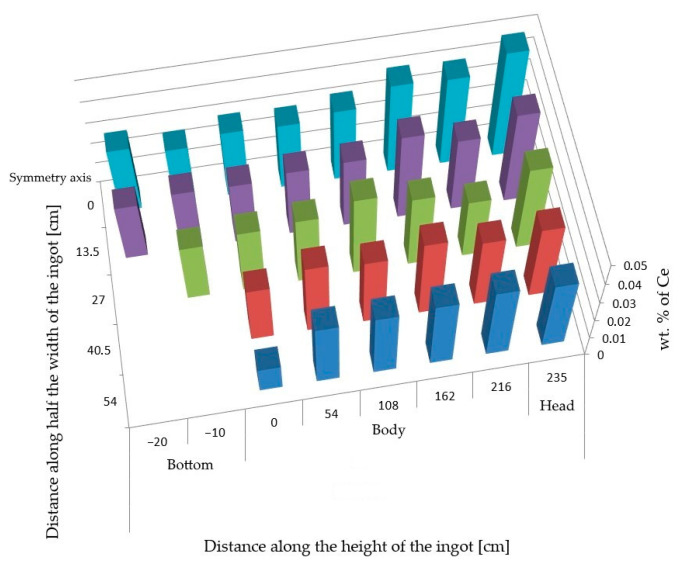
Distribution of cerium on ¼ of experimental ingot from heat A.

**Figure 3 materials-14-05160-f003:**
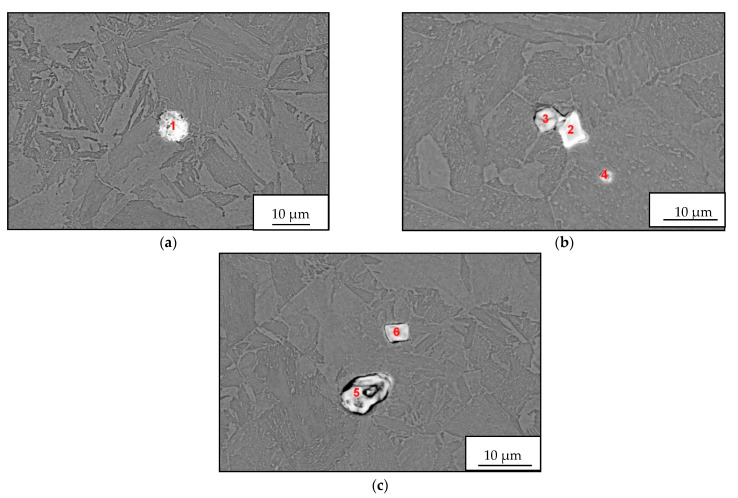
EDS microanalysis of REM inclusions detected in heat A, (BEC). (**a**–**c**) Chemical composition of oxides or oxisulfides corresponds to Spectrum 1–6 in Table 4.

**Figure 4 materials-14-05160-f004:**
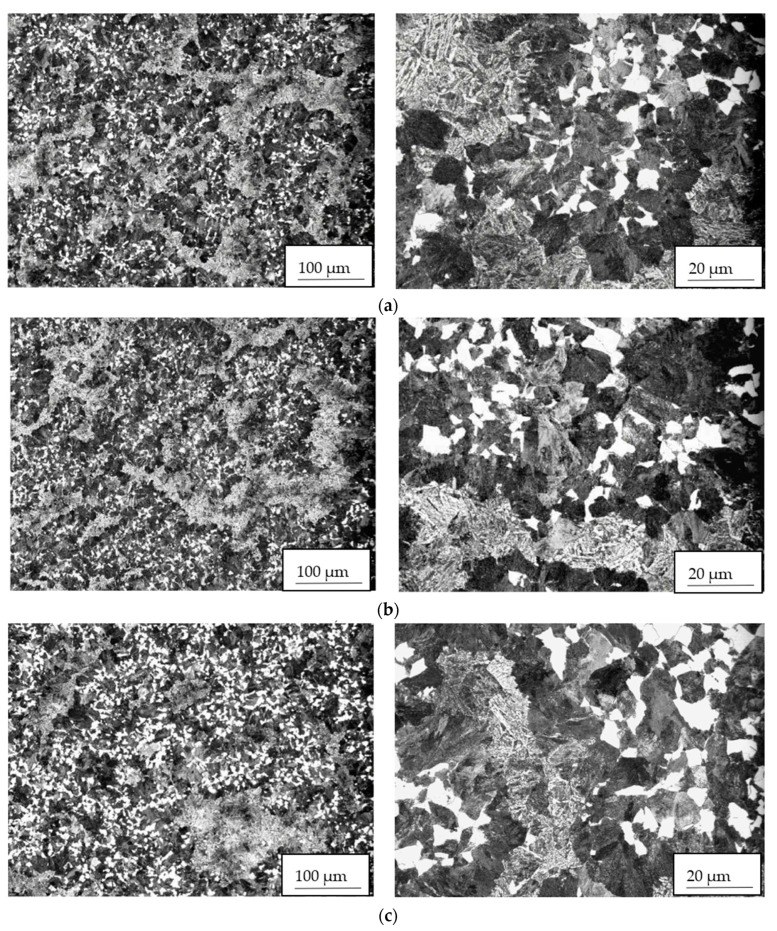
Microstructure formed by blocks of bainite, ferrite grains, and perlite nodules in cross-section of the studied steel forging, heat A: (**a**) subsurface; (**b**) ¼ of diameter; (**c**) center.

**Figure 5 materials-14-05160-f005:**
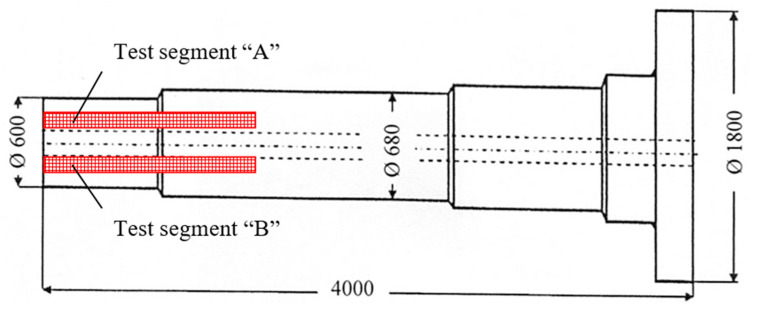
A sketch of forging made from a trial heat B. Areas with occurrence of unacceptable indications are marked in red.

**Figure 6 materials-14-05160-f006:**
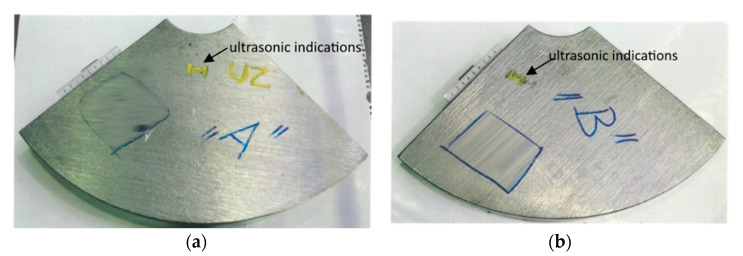
Test segments with detected ultrasonic indications sampled from forging B: (**a**) test segment “A”; (**b**) test segment “B”.

**Figure 7 materials-14-05160-f007:**
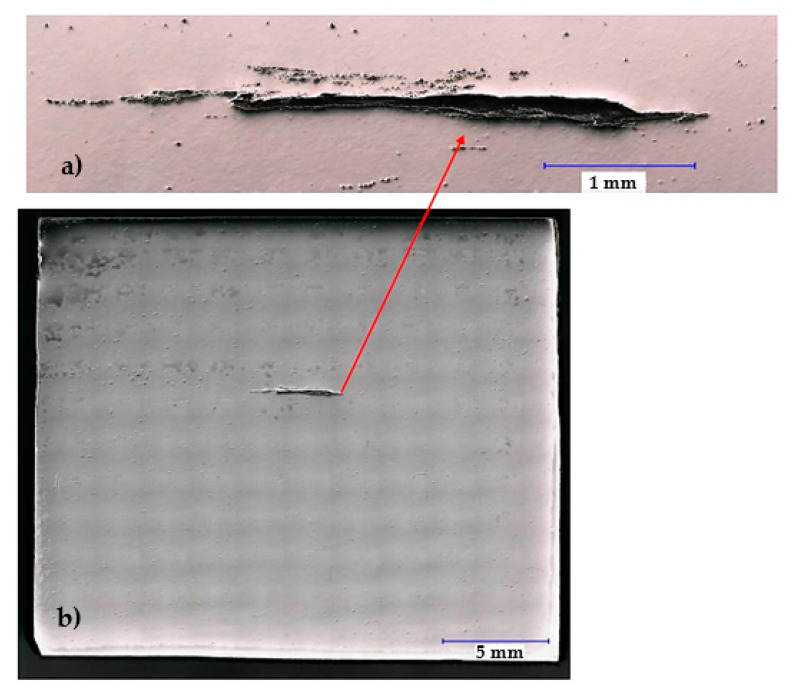
General view of sample 34/A with a crack of about 4.5 mm in length (**b**) Detail of crack surrounded by inclusions (**a**); Heat B.

**Figure 8 materials-14-05160-f008:**
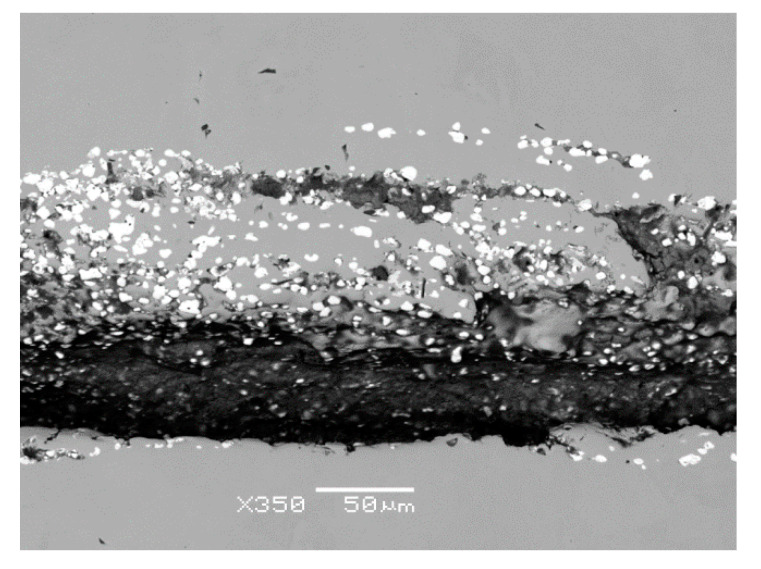
Row-oriented, nonmetallic inclusions, or clusters of nonmetallic inclusions inside or in the immediate vicinity of the crack, sample 34/A, heat B.

**Figure 9 materials-14-05160-f009:**
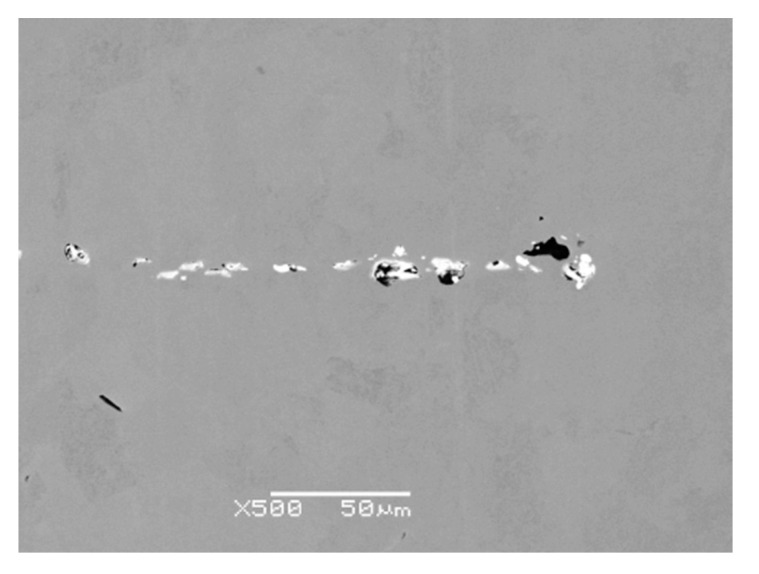
Nonmetallic oxide and oxysulfide inclusions in matrix, sample 34/A. Heat B.

**Figure 10 materials-14-05160-f010:**
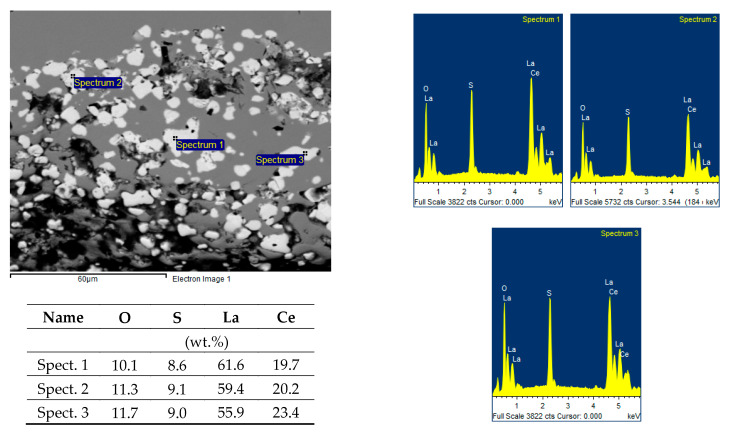
EDX spectrum of complex oxides, or lanthanum and cerium oxysulphides, sample 34/A. Heat B.

**Figure 11 materials-14-05160-f011:**
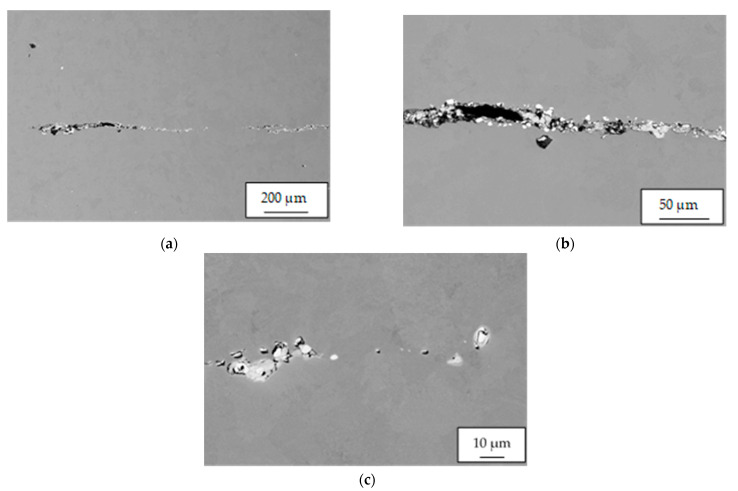
Sample 34/B: (**a**) crack and interconnected complex oxides or oxysulfide inclusions of REM forming small chains. Sample 34/B. Heat B; (**b**) complex oxides based on lanthanum and cerium with variable amounts of calcium and silicon, or lanthanum and cerium oxysulphides filling crack, sample 34/B. Heat B; (**c**) detail of chain of nonmetallic Ce/La inclusions.

**Figure 12 materials-14-05160-f012:**
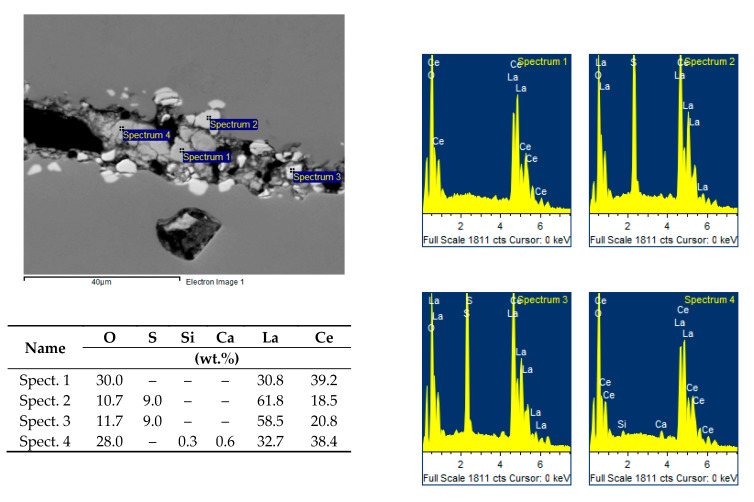
EDX spectrum of complex oxides based on lanthanum and cerium with variable amounts of calcium and silicon, or lanthanum and cerium oxysulphides, sample 34/B. Heat B.

**Figure 13 materials-14-05160-f013:**
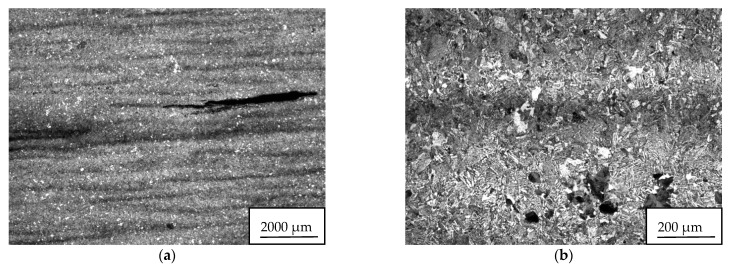
Microstructural characteristics of sample 34/A. Heat B: (**a**) quenched and tempered microstructure around crack; segregation bands of darker contrast correspond with the forming direction. (**b**) Detail of quenched and tempered microstructure consists of bainite, small islands of ferrite, perlite; segregation band of darker contrast is located in the center of the figure. (**c**) Detail of tempered bainite, ferrite (white islands), perlite (black areas); (**d**) detail of the microstructure in the segregation band; (**e**) surroundings of cracks; detail of quenched and tempered microstructure consists of bainite, small white islands of ferrite, black areas of perlite.

**Table 1 materials-14-05160-t001:** Chemical composition of steel after vacuum degassing, before and after the addition of mischmetal, heat A [wt.%]. * Oxygen activity was measured by CELOX^®^, hydrogen by HYDRIS^®^.

Before/After REM Add	C	Mn	Si	P	S	Cu	Ni	Cr	Mo	Al	Ce	H *	T [°C]	ao *
Before	0.42	0.69	0.26	0.008	0.001	0.13	0.19	1.02	0.16	0.023	–	0.000094	1596	0.000385
After	0.42	0.70	0.26	0.007	0.001	0.13	0.19	1.03	0.16	0.024	0.035	N/A	1573	0.000120

**Table 2 materials-14-05160-t002:** Chemical composition of slag during production of heat A [wt.%].

Slag Analysis	CaO	Al_2_O_3_	FeO	MnO	MgO	SiO_2_	Cr_2_O_3_	P_2_O_5_	S	TiO_2_	Ce
LF	60.54	22.71	0.70	0.066	8.370	7.17	0.020	0.015	0.32	0.110	–
VD before REM added	59.24	24.02	0.34	0.090	7.390	8.32	0.090	0.011	0.32	0.150	0.10
VD after REM added	69.69	22.83	0.45	0.071	8.100	7.42	0.018	0.011	0.33	0.110	1.00

**Table 3 materials-14-05160-t003:** Results of control chemical analysis of bar forged from heat A [wt.%].

Sampling Location	C	Mn	Si	P	S	Cu	Ni	Cr	Mo	Al	Ce	N	O
Subsurface	0.435	0.67	0.257	0.0060	0.0050	0.138	0.199	1.02	0.147	0.024	0.023	0.00622	0.00348
¼ of diameter	0.446	0.67	0.259	0.0060	0.0056	0.141	0.201	1.03	0.148	0.023	0.027	0.00589	0.00113
Center	0.454	0.69	0.285	0.0063	0.0057	0.135	0.127	1.03	0.157	0.025	0.028	0.00743	0.00181

**Table 4 materials-14-05160-t004:** EDS inclusion microanalysis and area chemical analysis of sample (average of 3 measurements) of heat A [wt.%].

Name	O	Si	P	S	Ca	Cr	Mn	Fe	Cu	As	Sb	La	Ce
Spectrum 1	11.22	-	9.68	0.35	-	-	-	3.41	0.37	26.00	5.81	-	43.16
Spectrum 2	7.49	0.18	1.51	-	-	-	-	3.07	-	34.19	5.80	19.84	27.92
Spectrum 3	26.22	0.28	-	-	-	-	-	8.55	0.20	-	-	-	64.75
Spectrum 4	21.31	-	-	0.89	-	-	-	16.48	-	-	-	6.71	54.61
Spectrum 5	31.27	0.20	-	-	0.35	-	-	4.25	0.23	-	-	15.80	47.90
Spectrum 6	26.00		-	-	-	-	-	8.93	0.24	-	-	10.57	54.26
Area analysis of the sample	-	0.41	-	-	-	1.24	0.78	97.57	-	-	-	-	-

**Table 5 materials-14-05160-t005:** EDX phase microanalysis in inclusions [at.%].

No.	Inclusions	O	Mg	Al	Si	P	S	Ca	Fe	Cu	As	Sb	La	Ce
1	(La-Ce)_x_(O_y_S_z_)∙SiO_2_	54.64			0.30		11.92		7.52	0.19			11.38	14.05
2	(La-Ce)_x_(O_y_P_z_)(As_v_Sb_w_)∙FeO	36.20				9.01			19.02	0.48	13.07	1.69	6.77	13.76
3	(La-Ce)_x_O_y_∙Al_2_O_3_∙MgO∙SiO_2_	60.52	1.78	17.03	0.68				6.68				2.28	11.03
4	(La-Ce)_x_(O_y_S_z_)∙FeO∙CaO∙SiO_2_	47.60			0.40		8.82	1.71	18.56				13.51	9.40

Note: Based on 23 analyses for Nr. 1, 7 for Nr. 2, 4 for Nr. 3, 13 for Nr. 4.

**Table 6 materials-14-05160-t006:** Mechanical tests results of the studied steel forging (0.035 wt.% Ce), heat A.

Sampling Location	0.2% OYS (MPa)	TS (MPa)	Elongation, 5D (%)	Contraction (%)	Charpy V-Notch Impact Energy, +20 °C (J)
Subsurface	568	794	17.8	49	40; 42; 38
¼ of diameter	479	790	16.5	41	34; 12; 12
center	418	751	17.0	39.5	12; 12; 28

**Table 7 materials-14-05160-t007:** Chemical composition before and after the addition of mischmetal to VD, heat B [wt.%]. * Oxygen activity was measured by CELOX^®^, hydrogen by HYDRIS^®^.

Before/After REM Add	C	Mn	Si	P	S	Cu	Ni	Cr	Mo	Al	Ce	H *	T (°C)	ao *
Before	0.40	0.74	0.24	0.009	0.002	0.10	0.56	1.10	0.25	0.026	–	0.000074	1566	0.00012
After	0.40	0.76	0.24	0.009	0.001	0.10	0.57	1.10	0.26	0.028	0.064	N/A	1556	N/A

**Table 8 materials-14-05160-t008:** Results of control chemical analysis of shaft forged from heat B [wt.%].

C	Mn	Si	P	S	Cu	Ni	Cr	Mo	Al (Dis.)	Al (Bond.)	Al (Total)	Ce
0.46	0.76	0.20	0.015	0.001	0.092	0.57	1.10	0.26	0.024	<0.001	0.024	0.055

## Data Availability

Not applicable.

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
