# Peer review of "The Effect of Rare Earth Metals Alloying on the Internal Quality of Industrially Produced Heavy Steel Forgings"

_materials, 2021, doi:10.3390/ma14185160_

Round 1

Reviewer 1 Report

This paper is the experimental and metallurgical study on the effect of REM on the large steel ingot to be forged.  
Because the reviewer is based on macroscopic point of view, I could not give some meaningful advices.

English should be improved. The concise and dry expression is better than fancy and complex expression in engieering writing. Please use simpler English. Followings would rather be checked:
-from point of view better mechanical properties of forging
-lead to cracks
-The studied steels were quenched and high-temperature tempered and consisted of 24 tempered bainite, ferrite and perlite. I think the 'microstructure' could not be quenched.
​​
What does 'The forging reduction was 5' mean?

Company names of the measuring devices employed do not guarantee the accuracy and the validity. They may look commercial. Please use some related references and hide them.

Some ambigous expression should be minimized. For example, probably.

The closing comments in the Conclusion should be minimized, which can damage the new findings.

Author Response

Dear Reviewer,

Thank you for reading our manuscript and reviewing it. Your comments and suggestions help us to improve our manuscript to a better scientific level. We revised our manuscript, all changes are in red.

Thank you again for your time and kindness.

Please find below the point-by-point answers:

English should be improved. The concise and dry expression is better than fancy and complex expression in engieering writing. Please use simpler English. Followings would rather be checked:
-from point of view better mechanical properties of forging
-lead to cracks
-The studied steels were quenched and high-temperature tempered and consisted of 24 tempered bainite, ferrite and perlite. I think the 'microstructure' could not be quenched.
corrected according to the suggestions

What does 'The forging reduction was 5' mean?

It is means ratio of the cross-sectional areas before and after forging (lengthened of the piece between dies)

Company names of the measuring devices employed do not guarantee the accuracy and the validity. They may look commercial. Please use some related references and hide them.

We agree but at previous published paper the manufacturer of experimental devices was required by MDPI Editorial Office. We removed the names

Some ambigous expression should be minimized. For example, probably.

I think, expressions e.g. probably, it seems, etc.… are commonly used in technical practice by many researchers. Each research and development contents an smaller or bigger uncertainty.

The closing comments in the Conclusion should be minimized, which can damage the new findings.

We have shortened the Conclusions’ text

Reviewer 2 Report

Dear Editor: I would like to express my deep thanks for inviting me to review the manuscript ID: materials-1286213

Title:    The Effect of Rare Earth Metals Alloying on the Internal Quality of Industrially Produced Heavy Steel Forgings

Authors: Petr Jonšta, ZdenÄ›k Jonšta, Silvie Brožová, Manuela Ingaldi, Jacek Pietraszek and Dorota Klimecka-Tatar

Comments:

Abstract:

Please delete this sentence “The aim of the paper was to verify the newly designed steelmaking process using Rare 13 Earth Metals (REMs) in the production of high-quality heavy ingots from the 42CrMo4 structural steel and to assess the effect of REMs alloying on microcleanness, microstructural and mechanical 15 properties of forgings”

Please rewrite the abstract according to your results.

Introduction:

There add recent relevant research work. Please rewrite the introduction section.

Clearly explain the objectives and novelty.

Results and discussion:

  • Please explain details in Table 2 VD after REM added why Ce content is too high in comparison with other process.
  • According to Figure 2, it is clear that Ce content is not uniform throughout the specimens, explain the reason
  • Figure 3 is not clear, provide clear images after etching?
  • Microstructures formed by blocks of bainite, ferrite grains and perlite nodules in Figure 4 are not clear. Please provide EBSD image.
  • Please combine Figure 11, 12 and 13
  • Figure 15 does not provide any information. Please provide clear images.

Conclusion part:

Please concise the conclusion parts.

RECOMMENDATION

After reviewing the enclosed manuscript for “Materials”, the present manuscript contains some kinds of scientific analysis but it is mandatory required to modify according to the preceding remarks. So, the manuscript can be publication after major revision.

Author Response

Dear Reviewer,

Thank you for reading our manuscript and reviewing it. Your comments and suggestions help us to improve our manuscript to a better scientific level. We revised our manuscript, all changes are in red.

Thank you again for your time and kindness.

Please find below the point-by-point answers:

Comments:

Abstract:

Please delete this sentence “The aim of the paper was to verify the newly designed steelmaking process using Rare 13 Earth Metals (REMs) in the production of high-quality heavy ingots from the 42CrMo4 structural steel and to assess the effect of REMs alloying on microcleanness, microstructural and mechanical 15 properties of forgings”

We deleted this sentence.

Please rewrite the abstract according to your results.

Abstract has been rewritten.

Introduction:

There add recent relevant research work. Please rewrite the introduction section.

Knowledge in the technical literature regarding REM alloying for heavy ingots in the order of tens of tons is not commonly available. In addition, concept of REM alloying is also mostly studied for high alloyed materials. The Introduction section has been revised. Developed by some information from the latest publications

Clearly explain the objectives and novelty.

This part has been corrected.

Results and discussion:

  • Please explain details in Table 2 VD after REM added why Ce content is too high in comparison with other process.

Thank you for the question. This is not clear and it is surprising. Additional analysis would be very useful. The slag adhere to the lid of the ladle may have played a role. I suggest a change in text, please.

  • According to Figure 2, it is clear that Ce content is not uniform throughout the specimens, explain the reason.

In general, chemical non-uniformity is related to segregation processes during solidification and cooling of the heavy steel ingot. It depends on many factors (temperature gradient, solidification time, shape of mould, etc.)

  • Figure 3 is not clear, provide clear images after etching?

The purpose of figure is present kind of nonmetallic inclusions. Subsequently, their compositions are presented in Tables 4, 5 and discussed in text.

  • Microstructures formed by blocks of bainite, ferrite grains and perlite nodules in Figure 4 are not clear. Please provide EBSD image.

I am sorry but EBSD technique was not available.

  • Please combine Figure 11, 12 and 13

Corrected

  • Figure 15 does not provide any information. Please provide clear images.

The figures showed defects detected by ultrasonic testing, documentation was performed by standard techniques, commonly used. Unfortunately, add other figures or change actually is not possible. Thank you very much for understanding.

Conclusion part:

Please concise the conclusion parts.

Corrected

Reviewer 3 Report

1. The use of which elements will improve the quality of the rock steel?  

2. What level of properties would you like to achieve in these materials?  

3. Is it advisable to use REM?

Author Response

Dear Reviewer,

Thank you for reading our manuscript and reviewing it. Your comments and suggestions help us to improve our manuscript to a better scientific level. We revised our manuscript, all changes are in red.

Thank you again for your time and kindness.

Please find below the point-by-point answers:

  1. The use of which elements will improve the quality of the rock steel?

I am not sure I understand the question correctly. If you mean geology, I am sorry, it is out of my area of professional interests.

  1. What level of properties would you like to achieve in these materials?

Continuously increasing power output of wind power plants, respectively turbines leads to increasing requirements of mechanical properties of rotor shafts. It is also important to control level of properties in the different areas in the cross section of forgings, i.e. „properties homogeneity” and good repeatability of production. Specific values of mechanical properties are listed in technical documentation of customers and cannot be published openly.

  1. Is it advisable to use REM?

It is one possible way to try to improve structural parameters and achieve better mechanical properties. Under certain conditions can lead to production costs reduction (e.g. reduction of the number of repeated heat treatments of forgings with unsatisfactory mechanical properties due to non-fine grained structure).

Round 2

Reviewer 2 Report

Authors addressed all comments in the revised manuscript.

Reviewer 3 Report

I was satisfied with the changes made by the authors